# Overexpression of *T-bet*, *GATA-3* and *TGF-ß* Induces *IFN-γ*, *IL-4/13A*, and *IL-17A* Expression in Atlantic Salmon

**DOI:** 10.3390/biology9040082

**Published:** 2020-04-20

**Authors:** Tiril H. Slettjord, Hege J. Sekkenes, Heng Chi, Jarl Bøgwald, Trilochan Swain, Roy A. Dalmo, Jaya Kumari Swain

**Affiliations:** 1Norwegian College of Fishery Science, Faculty of Biosciences, Fisheries and Economics, University of Tromsø - The Artic University of Norway, N-9037 Tromsø, Norway; tiril.slettjord@cermaq.com (T.H.S.); hege@sekkenes.com (H.J.S.); chiheng@ouc.edu.cn (H.C.); jarl.bogwald@uit.no (J.B.); trilochan.swain@uit.no (T.S.); 2Cermaq Ltd., Gjærbakknes, N-8286 Steigen, Norway; 3Val FoU AS, Val, N-7970 Kolvereid, Norway; 4Key Laboratory of Experimental Marine Biology, Institute of Oceanology, Chinese Academy of Sciences, 7 Nanhai Road, Qingdao 266071, China; 5NOFIMA, Muninbakken 9-13, N-9291 Tromsø, Norway

**Keywords:** overexpression, *T-bet*, *GATA-3*, *TGF-ß*, Atlantic salmon, gene expression

## Abstract

The overexpression of *GATA-3*, *T-bet* and *TGF-ß* may theoretically induce *IL-4/A*, *IFN-γ* and *IL-17A* expression, respectively. Whether this also applies to fish is not yet known. The plasmid vectors encoding reporter gene (RFP)-tagged *T-bet*, *GATA-3* and *TGF-ß* were used as overexpression tools, transfected into cells or injected intramuscularly to monitor the expression of *IFN-γ*, *IL-4/13A* and *IL-17A*. In addition, the fish were either experimentally challenged with *Vibrio anguillarum* (VA group) or *Piscirickettsia salmonis* (PS group). The reporter gene (RFP) inserted upstream of the *GATA-3*, *T-bet* and *TGF-ß* genes, was observed in muscle cell nuclei and in inflammatory cells after intramuscular (i.m.) injection. PS group: following the injection of *GATA-3* and *T-bet*-encoding plasmids, the expression of *GATA-3* and *T-bet* was high at the injection site. The spleen expression of *IFN-γ*, following the injection of a *T-bet*-encoding plasmid, was significantly higher on day 2. VA group: The *T-bet* and *GATA-3*-overexpressing fish expressed high *T-bet* and *GATA-3* mRNA levels in the muscles and on day 4 post-challenge. The expression of *TGF-ß* in the muscles of fish injected with *TGF-ß*-encoding plasmids was significantly higher on days 7 (8 days pre-challenge) and 19 (4 days after challenge). The protective effects of the overexpression of *T-bet*, *GATA-3* and *TGF-ß* on both bacterial infections were negligible.

## 1. Introduction

The aquaculture production of Atlantic salmon is threatened by infections from a wide array of pathogens. Some of them are not effectively controlled by current vaccines (consisting of inactivated pathogens in water-in-oil formulations), which mainly show effects through their B-cell response, whereas a robust T cell response is suboptimal, thus limiting the vaccine efficacy. One way to increase vaccine efficacy is to develop targeting molecular adjuvants [1,2,3], which may induce T cell activation. It is unknown whether the T cell differentiation process in fish is equal to that described in higher vertebrates and mammalian species.

There are several developmental checkpoints during T cell differentiation, where regulation by a combination of transcription factors imprint specific functional properties on precursors [4]. Two of these are *T-bet* and *GATA-3*, which govern naïve T helper cells to differentiate between Th1 and Th2 cells, depending on the local cytokine production by dendritic cells and macrophages, respectively [5]. It is generally accepted that Th1 cells produce interferon-γ (*IFN-γ*), which promotes cell-mediated immunity, whereas Th2 cells may control infections from extracellular pathogens with the presence of (mainly) IL-4 and IL-13 [6]. *TGF-ß* may be important for the differentiation of Th cells into a regulatory Th or/and Th17 subtype, and may control the amplitude of an inflammatory response (Treg) and/or induce inflammation (Th17) [7]. There may be a certain degree of plasticity between the different Th subtypes, depending on the local cytokine milieu and which transcription factor is activated. It has been proposed that the Th17 response is involved in the protection against certain Gram-negative bacteria, such as *Klebsiella pneumonia* infection [8].

It is widely acknowledged that a Th1-like response may confer protection against pathology and disease caused by intracellular bacteria [9]. For example, an infection study on mice has shown that *T-bet* deficiency (*T-bet*
^-/-^ mice) lead to increased *Mycobacterium tuberculosis* susceptibility, indicating that *T-bet* has a central function in the regulation of cytokine production (*IFN-γ* and IL-10) and in controlling the infection. A similar impact has also been described on other infectious diseases caused by bacteria [9]. *T-bet* knockouts have severe defects in Th1 cell differentiation, both in vitro and in vivo. *IFN-γ* responses to *Leishmania major* are significantly down regulated in *T-bet*
^−/−^ mice. These mice showed increased IL-4 and IL-5 production (Th2 cytokines) in response to *Leishmania major* infection [10].

Infection studies using *GATA-3*
^-/-^ mice have not been possible, as the germline deletion of *GATA-3* often results in embryonic lethality [11]. However, by using conditionally *GATA-3* deficient mice, studies have shown that these mice are prompted to display a shift towards a Th1 response after immunization (with alum). Thus, the importance of *GATA-3* for initiating and maintaining Th2 responses while regulating Th1 responses has been documented with *GATA-3*-deleted CD4^+^ T cells [12]. Conditionally, *GATA-3*-deficient mice have also been reported to be more susceptible to *Citrobacter rodentium* infection (a Gram-negative bacterium that infects mostly mucosal tissues) [13,14]. Transgenic mice that overexpress *GATA-3* have been developed and are reported to be more vulnerable to *Candida albicans* infection, suggesting that the overexpression of *GATA-3* induces a shift to a Th2 response during infection [15].

Regulatory Foxp3^+^ T cells (Foxp3^+^ Treg) may act as chief regulators of homeostasis, where their action minimizes tissue destruction during inflammation [16]. It has been suggested that *TGF-ß* is one of the main cytokines for the induction and regulation of Treg activity [17]. However, during inflammation or infection, natural Treg cells may show plasticity and become IL-17 and *IFN-γ*-producing *T-bet*^+^ Treg cells (Th17-like Treg) [18]. Being a pleiotropic molecule, *TGF-ß* is also essential for the generation of Th17 and Th9 cells [19]. In addition, it has been suggested that *TGF-ß* plays a role in Th17 differentiation by suppressing the expression of *T-bet* and *GATA-3* [20].

Whether fish possess an equal or similar form of T cell differentiation and activities, in a similar way to mammalian species, remains unknown. The adaptive immune system of fish, in general, resembles what is present in higher vertebrates, though differences exist [21,22]. One of the major challenges in understanding their adaptive immune responses is that salmonid fish species produce, in many instances, several paralogs/subtypes of immune molecules due to their partial tetrapoloidy [23,24]. This also applies to T cell biology [25,26,27].

*Piscirickettsia salmonis* is an intracellular pathogen [28], whereas *Vibrio anguillarum* is a extracellular pathogen [29]; both pose potential threats to salmonid aquaculture. Whether any ectopic expression of *T-bet*, *GATA-3* or *TGF-ß* induces disease resistance against *V. anguillarum* and *P. rickettsia* has not previously been evaluated.

In this study, we hypothesize that the plasmid DNA-induced overexpression of *T-bet*, *GATA-3* and *TGF-ß* induces Th1, Th2 and Th17-like responses in Atlantic salmon and that this overexpression might modulate the level of protection against an experimental challenge with intracellular (*P. salmonis*) and extracellular (*V. anguillarum*) bacteria. The overexpression of genes that are important during T cell differentiation has never been performed in fish; such overexpression studies may provide a novel tool to study T cell biology in fish in general.

## 2. Material and Methods

### 2.1. Fish

Atlantic salmon (*Salmo salar* L.) presmolts (30–50 g) of the strain “AquaGen standard” (AquaGen AS, Kyrksæterøra, Norway) were kept at Tromsø Aquaculture Research Station, Norway, in 300 L tanks, continuously supplied with fresh water at 14 °C and were fed commercial dry food (Skretting Olympic 2.0 mm, Skretting AS, Stavanger, Norway). Immediately before treatment, the fish were anesthetized with 0.005% benzocaine (ACD Pharmaceuticals, Oslo, Norway). Fish were tagged by Panjet tattooing using 2% Alcian blue. An overdose of benzocaine prior to the harvesting of the organs killed the fish. The Norwegian Animal Research Authority (NARA) approved all experimental protocols involving live fish, in compliance with the Animal Welfare Act (https://www.regjeringen.no/en/dokumenter/animal-welfare-act/id571188/) (approval nos. 2012/5248 and 11/247913). We confirm that all experiments were performed in accordance with the relevant guidelines and regulations outlined by the Norwegian Animal Research Authority.

### 2.2. Plasmid Construction Used for Intramuscular Injection

For the construction of plasmid-encoding *TGF-ß*, *T-bet* and *GATA-3* genes with the reporter gene (*RFP)*-Tag, the open reading frames (ORFs) of salmon *TGF-ß* gene (BT059581), *T-bet* (GU979861) and *GATA-3* (EU418015) were all retrieved from the GenBank and cloned, as described by Kumari et al. (2009, 2015) [30,31]. From the spleen cDNA library, the genes were sub-cloned into a *pTagRFP-N* vector (Evrogen) by PCR, with the aid of restriction enzymes SacII and Xho I. All the PCR products of *T-bet*, *GATA-3* and *TGF-ß* genes and the *pTagRFP-N* vector were digested with endonuclease (New England Biolabs, MA, USA) and ligated (T4 DNA ligase) to generate the mentioned constructs. A re-ligated *pTagRFP-N* plasmid without insertions was constructed and used as the negative control. All plasmids were transformed and grown in One Shot TOPO 10 *Escherichia coli* (Invitrogen, Carlsbad, CA, USA), isolated using a Plasmid Mini Kit (Qiagen, Hilden, Germany), then verified through a restriction map analysis and DNA sequencing. The purification of plasmids was carried out using the Qiagen Plasmid mega (for the pilot study) or Qiagen Giga kit, according to manufacturer’s protocols. The anion exchange chromatography-purified plasmid DNA was further diluted in phosphate buffered saline (PBS) (pH 7.4) (Sigma, St. Luis, MO, USA) before injection into the fish.

### 2.3. Cell Culture and Transfection Assay

Chinook salmon embryonic cells (CHSE-214) were seeded in standard flasks (Nunc™) with L-15 medium (Invitrogen) containing penicillin (60 μg ml^−1^), streptomycin (100 μg ml^−1^), 1% non-essential amino acids (NEAA, Gibco) and 8% fetal bovine serum (FBS). The CHSE-214 cells were incubated at 20 °C for 1 week. Cells were then washed twice with 10 mL PBS before adding 1.5 mL trypsin (Sigma). Loosened cells were re-suspended in L-15 medium (8% FBS, 1% NEAA, without antibiotics) and counted using a Nucleocounter YC-100 (Chemometec, Allerod, Denmark). Transfections of plasmids were performed using the Invitrogen Neon^TM^ Transfection System according to the supplier’s protocols (Invitrogen). The cells were either transfected with *pTagRFP-T-bet*, *pTagRFP-GATA-3, pTagRFP-TGF-β* or pTagRFP-N, expressing plasmids in separate wells (1 × 10^5^ cells per well). Mock-transfected cells were used as the controls. After 12 h of incubation in the transfection medium, the cells were gently washed and received L-15. After 48 h, the cells were fixed with 4% formaldehyde (*w*/*v*) (Thermo Scientific, Waltham, MA, USA) for 30 min, then DAPI (4′,6-diamidino-2-phenylindole ) (Sigma) was used for nucleic acid staining, according to the manufacturer’s protocols. Micrographs were obtained with an inverted fluorescence microscope equipped with DAPI-365 and Texas Red 530-585 filters (Zeiss Axiovert 40 CFL).

### 2.4. Injection of Plasmid DNA, Experimental Challenge and Sampling

Before the main experiment was undertaken, a pilot experiment was performed. In this pilot experiment, fish were injected with 70 µg (2.3 mg kg^−1^ body weight) pDNA (*pTagRFP-T-bet*, *pTagRFP-GATA-3* or *pTagRFP-TGF-β*) and 50 µL PBS. Subsequent qPCR analyses of the expression dynamics of *T-bet* and *GATA-3* were undertaken to find a suitable protocol for the main experiment. Likewise, a pilot challenge experiment was performed to find the bacterial median lethal dose (LD50) dose that resulted in mortalities. Throughout the main study, experimental fish were kept in fresh water at 14 °C with a water flow of 1 L min^−1^ to 300 L tanks. Pre-smolts of Atlantic salmon weighing approximately 30 g were injected intramuscularly (i.m.) with 20 µg (667 µg kg^−1^ body weight) (amount deduced from the pilot trial) of plasmids diluted in 50 µL PBS per fish. *T-bet*, *GATA-3* and *TGF-ß*-encoding plasmids were used for overexpression studies, whereas the two groups of control fish received control plasmid and PBS, respectively. Fifteen fish were sacrificed before the onset of the injection experiment; samples from these fish were used as non-treated zero-point samples. Subsequently, a total of 1065 fish were separated in five different treatment groups with 210 fish in each and tagged (Table 1). After tagging and recovery, 120 fish from each treatment group were evenly divided into six parallel tanks; these six tanks (300 L) were left undisturbed until the onset of the challenge experiment. In addition, 90 fish from each treatment group were transferred evenly to two parallel tanks—these tanks contained fish for sampling purposes only. Four hundred and eighty fish (80 per treatment) from the undisturbed tanks were challenged by *P. salmonis* (day 15), whereas the same number of fish were injected with *V. anguillarum* (day 15). A total of 96 fish were used for sampling before the start of the bacterial challenge; the remaining fish (264 animals) were divided evenly and also subjected to bacterial infection with *V. anguillarum* and *P. salmonis*, respectively (Figure 1). These fish were transferred to two tanks and subjected to sampling. As there was a very high rate of mortality in the *V. anguillarum*-infected fish, the fish meant for sampling at time points beyond day 19 were lost from the experiment. The challenge experiment for each pathogen (undisturbed tanks) was performed in triplicate.

As mentioned above, the fish were separated into two subgroups: *V. anguillarum* (VA) and *P. salmonis* (PS) subgroups (Figure 1). Fish in one sub-group were injected intraperitoneally (i.p.) with 100 μL of *Vibrio anguillarum* (O2a isolate, LFI 323) (nine colony forming units of live bacteria per 100 µl PBS), while, in the other, fish were injected with 100 μL of *P. salmonis* (dose restricted, to be published by Pharmaq, Oslo, Norway), respectively. From the *V. anguillarum* subgroup, only one sampling, four days (day 19) after infection, was possible due to the high mortality rate, while in the experimental challenge with *P. salmonis,* sampling was undertaken at day 4, 7 and 13 post-infection (Figure 1). Mortality was recorded every day until 19 days post-bacterial infection, and dead fish were removed from tanks daily. Post-infection, sampling for qPCR analysis was performed only for live or moribund fish, not for dead fish. The relative survival percentage (RSP) was calculated using the following formula: (1- (number of deaths/total)) × 100) (modified by Costa et al. (2011)) [32]. Furthermore, the levels of *V. anguillarum* and *P. salmonis* in 16S rRNA transcripts were analyzed by qPCR in the head kidney before and after bacterial infection.

### 2.5. Sampling for qPCR

Sampling (six fish per treatment group per time point; N = 6) was performed for plasmid DNA and PBS-administered fish 2- and 7-days post-injection (Figure 1). cDNA samples from different replicate samples were not mixed. Samples were also obtained from infected fish at day 19 (both *V. anguillarum* and *P. salmonis* infected fish), and on day 22 and 28 post-injection for *P. salmonis*-infected fish only (from the head kidney, spleen and muscle tissues) and were stored in RNA, according to the manufacturer´s protocols, for gene expression studies at a later date.

### 2.6. Reverse Transcription and Quantitative PCR (qPCR)

RNA isolation and cDNA synthesis were performed according to Kumari et al. (2015) [31]. The synthesized cDNA was diluted 10-fold with MilliQ water, and 5 μL of this dilution was used as a template in a 20 μL reaction volume. qPCR was performed in triplicate, as described in Kumari et al. (2013) [33]. In all cases, amplifications were specific, and no amplification was observed in negative controls (non-template control and non-reverse transcriptase control). The Ct values for each sample were converted into fold differences according to the ΔΔCT method [34] for relative quantitation, since the PCR efficiencies between the targets and endogenous control were relatively equivalent. Elongation factor 1α (*EF-1α*) was found to be the most stable reference gene compared to *18S* rRNA. Therefore, gene expression was normalized by EF-1α (timepoint controls) in each sample. The primers used are shown in Table 2.

### 2.7. Fluorescence Microscopy

At days 7, 14 and 21, spleen, head kidney and muscle samples from the injection site were dissected and fixed in 4% formalin for one week at room temperature (RT), then transferred to 70% ethanol for storage and further processing. The tissue and cellular localization of RFP-tagged protein expression was investigated using fluorescence microscopy. Tissue samples from fish that received PBS were used as controls. The fixed tissue samples were dehydrated and embedded in paraffin (Leica EG 115 OH), cooled (Axel Johnson CP-4 cooling plate), sectioned (4 μm), and de-waxed in xylene/HistoClear (Merck/National diagnostics, Kenilworth, NJ, USA), then mounted in Eukitt^®^ (O. Kindler, Freiburg, Germany). Standard contrast stains were omitted. For some specimens, DAPI staining (nuclei staining; light blue color) was performed to look for any co-localization of RFP and DAPI staining. RFP fluorescence was observed as a bright red color. Sections were examined and photographed using Leica DM 6000 B and CTR 6000 at 100× and 400× magnification. Fluorescence microscopy samples from the pilot experiments were also analyzed and showed similar RFP specific fluorescence compared to samples obtained from the main experiment.

### 2.8. Statistical Analysis

The log-transformed data were analyzed by one-way analysis of variance (ANOVA), followed by Tukey’s multiple-range test to determine the differences between groups using SYSTAT 13 software. Graphs were plotted using GraphPad Prism 7. Differences were considered statistically significant when *p* < 0.05.

## 3. Results

### 3.1. In Vitro Expression of *T-bet*, *GATA-3* and *TGF-ß*

To determine whether the expression vectors induced gene expression in vitro, and to gain information on the sub-cellular localization of *T-bet*, *GATA-3* and TGF-β expression following transfection, fluorescence microscopy of cells transfected with *T-bet*, *GATA-3* and TGF-β-encoding plasmids that were fused to red fluorescent protein (RFP) was carried out. Approximately 30–40% of the CHSE-214 cells expressed RFP-specific red staining. As shown in Figure 2, both the *T-bet* and *GATA-3* fused to the *RFP* gene were, in most instances, co-localized with DAPI (nucleic staining), whereas RFP-tagged *TGF-ß* was typically localized in the cytoplasm at 48 h post-transfection, though co-localization with DAPI was also observed. DAPI was used for nuclei counterstaining (blue) (Figure 2). Mock-transfected cells showed no staining.

### 3.2. Fluorescence Microscopical Detection of RFP In Vivo

To assess the in vivo expression of the RFP transgene fused to *T-bet*, *GATA-3* and *TGF-ß* genes, fluorescence microscopy of the muscle tissue sections (injection site) obtained from fish on days 2 and 7, after i.m. administration of the corresponding plasmids, was performed. The analysis revealed the presence of RFP-elicited fluorescence in all the sections obtained from fish that received *pTagRFP-T-bet*, *pTagRFP-GATA-3* and *pTagRFP-TGF-ß*. There were expressions of RFP from *T-bet* and *GATA-3*-encoding plasmids in a few muscle cell nuclei (Figure 3). No expression of *TGF-ß* from *pTagRFP-TGF-ß* was found in muscle cells (cytoplasm or nuclei). Interestingly, inflammatory cells surrounding the muscle cells contained red fluorescence (Figure 3) in samples from all groups (*pTagRFP-T-bet*, *pTagRFP-GATA-3* and *pTagRFP-TGF-ß*). Through DAPI staining, we hoped to distinguish the RFP fluorescence from cell nuclei staining, or to find any co-localization of RFP with DAPI. Indeed, co-localized DAPI and RFP staining could be observed, together with staining that did not overlap, suggesting the nuclear and cytoplasmic expression of the transgenes (Figure 3). The empty or null plasmid control, containing the RFP gene, did not induce fluorescence in the nuclei of muscle cells and in surrounding cells. The tissue sections from fish injected with PBS contained no fluorescence (not shown). The tissue section consisted of necrotized muscle cells and an influx of inflammatory cells that followed the needle’s trajectory. Minor fluorescence was observed in distances 200 µm from the needle trajectory. No RFP was observed in any other tissues or organs.

### 3.3. Gene Expression after Intramuscular Plasmid Injection and P. salmonis Challenge

Firstly, we were interested in analyzing whether an injection of *GATA-3*, *T-bet* and *TGF-ß* encoding plasmids induced overexpression. The expression of *GATA-3* and *T-bet* was significantly higher than the time-matched plasmid controls at all timepoints pre (day 2 and 7) and post-infection (day 19, 22 and 28) at the injection site. The highest expression was on day 7 post-i.m. injection with the *GATA-3*-encoding plasmid (*p* < 0.001), and on day 2 after injection of the *T-bet*-expressing plasmid (*p* < 0.001) (Figure 4A,B). Moreover, the expression of transcription factors *T-bet* and *GATA-3* in the hematopoietic organs, spleen and head kidney were not higher than in their time-matched controls (days 2–28). However, *TGF-ß* expression seemed to be increased on day 7 and, though not statistically significant, the expression levels were similar to the controls after experimental infection on day 15. In the hematopoietic organs, the expression levels of *TGF-ß* and the control plasmid were variable during the course of the experiment (Figure 4C).

Secondly, an overexpression of *GATA-3, T-bet* and *TGF-ß* may theoretically induce *IL-4/A, IFN-γ* and *IL-17A* expression, respectively, thus we ran qPCR to calculate the *IL-4/13A*, *IFN-γ* and *IL-17A* mRNA levels. In the muscle tissue, there was higher expression of *IL-4/13A* than in the time-matched plasmid controls on day 19 (*p* < 0.01) and 28 (*p* < 0.01), with the highest level occurring on day 7, which is in line with the highest expression of *GATA-3* in the muscles (Figure 4A). The expression of *IFN-γ* in the spleen, following the injection of the *T-bet*-encoding plasmid, was significantly high at day 2 compared to the controls, and was generally lower at all other timepoints (day 7, 19, 22 and 28) compared to corresponding values in the head kidney. The control plasmid induced significantly higher expression of *IFN-γ* in the head kidney on days 22 and 28 (7 and 13 days post-infection) compared to fish that received the *T-bet*-encoding plasmid (Figure 4B). The expression of *IL-17A* mRNA in the muscle samples was below the detection levels, thus the muscle values have been omitted. The expression levels of *IL-4/13A*, *IFN-γ* and *IL-17A* in the spleen and head kidney were, in general, highly variable (Figure 4A–C). Following the administration of *T-bet* and *TGF-ß*-encoding plasmids, the expression of *IFN-γ* and *IL-17A* were lower on day 22 and day 28 compared to the corresponding values of the control plasmid group.

### 3.4. Gene Expression Post-Intra-Muscular Plasmid Injection and V. anguillarum Challenge

Similarly, as described above, in the *T-bet* and *GATA-3*-encoding plasmid-injected fish, the expression of *T-bet* and *GATA-3* in the muscle was significantly higher (*p* < 0.01) at all time points post-injection (day 2 and 7) and on day 4 post-challenge compared to fish that received the control plasmid, albeit with gradually decreasing levels of expression over time (Figure 5A,B). The expression of *T-bet* and *GATA-3* was relatively low in the hematopoietic organs and was not regulated during the study period (Figure 5). Following overexpression by *GATA-3*, the expression of *IL-4/13A* in the spleen was significantly higher (*p* < 0.05) on day 19 (day 4 post-infection) relative to the controls.

Moreover, the expression of the *T-bet*-encoding plasmid followed by experimental challenge with *V. anguillarum* seemed to induce *IFN-γ* in the head kidney and spleen, though not in a statistically significant manner. The expression of *TGF-ß* in the muscles of fish injected with the *TGF-ß*-encoding plasmid was significantly higher (*p* < 0.05) at day 7 (8 days pre-challenge) and 19 (4 days after challenge) compared to the time-matched controls (Figure 5C).

## 4. Experimental Challenge Study

To test if *T-bet*, *GATA-3* or *TGF-ß* overexpression might contribute to any protective effect against the intra or extracellular pathogens of salmon, fish were challenged on day 15 with *P. salmonis or V. anguillarum* (Figure 1).

All the fish infected by *V. anguillarum* were dead by day 5 post-infection (Figure 6B). Therefore, we were only able to sample and analyze gene expression by qPCR at one timepoint post-infection (day 19). For the *P. salmonis* infected fish, disease symptoms started on days 6–7 and acute mortality appeared from day 13 post-infection. Only a few surviving fish were left in the tanks 14 days after infection, and no statistically significant different survival rates were found between the different plasmid-injected groups (Figure 6A).

Furthermore, to check if the overexpression of T cell factors had any effect on limiting the bacterial loads, we calculated the bacterial loads by means of qPCR, where the expression of *16S* rRNA was analyzed in head kidney tissues retrieved from fish at different time points post-challenge, and pre-challenge head kidney tissues (day 2 and 7) (Figure 7A,B).

The expression of *P. salmonis 16S* rRNA was negligible at day 2 and day 7 pre-infection (Figure 7A) in fish from the different groups. The trend showed that there was a continuous increase in *P. salmonis 16S* rRNA levels from day 4 until day 28 (day 13 post-infection, when the mortality was at its peak). On day 4 post-infection, the expression levels of *16S* rRNA in the head kidneys of fish that were injected with the non-encoding plasmid were statistically significantly higher (*p* < 0.05) than in the head kidneys of fish injected with PBS. On day 28, the expression levels of *16S* rRNA were more or less similar between different plasmid-injected groups (Figure 7A), whereas the expression of *16S* rRNA in fish kidneys that received pTagRFP-*T-bet* and the control and non-coding plasmids were statistically higher than in PBS-injected fish.

The relative expression of *V. anguillarum* 16S rRNA in the head kidney prior to infection was close to zero on days 2 and 7, whereas on day 19 (4 days post-infection) the expression of *16S* rRNA was highest (but not statistically significant) in fish that received the control plasmid (a 200-fold increase compared to pre-challenge values, approximately) and PBS-injected controls (a 150-fold increase, approximately) (Figure 7B). Moreover, the lowest expression of *16S* rRNA was found in *TGF-ß* (0.11-fold increase), followed by *GATA-3*-encoding plasmid-injected fish (30-fold increase), showing a reduction in the bacterial load, although not a significant reduction compared to the controls.

## 5. Discussion

A better knowledge of T cell responses in fish is highly beneficial for vaccine development, especially against hard-to-combat infectious diseases caused by intracellular pathogens. As an approach to studying the T cell response in salmon, we investigated the effect of intra-muscularly injected *T-bet*, *GATA-3* and *TGF-ß*-expressing vectors on *T-bet*, *GATA-3* and *TGF-ß* expression, and on Th1, Th2 and Th17-like responses and protection against intra or extracellular pathogens. This study is the first in its kind to analyze the effects that the overexpression of *TGF-ß* and other transcription factors and have on Th cytokine production in fish, and our method may provide a novel tool for the study of T cell biology in fish. In the current study, the insertion of T cell transcription factors and *TGF-ß* into the DNA plasmid vectors, followed by injection into the fish, facilitated, in general, gene expressions of *IL-4/13A*, *IFN-γ* and *IL-17A* at the site of injection (except *IL-17A*) and, on a few occasions, in the lymphoid tissues. These cytokines are reminiscent of Th1, Th2 and Th17-like responses in higher vertebrates. We cannot, however, offer functional proof of the presence of T cell subsets, by, e.g., FACS and Western blotting. Most recently, antibodies against salmonid CD4 and IL-4/13 have been developed [35,36], which would have been beneficial if they had been accessible during this study (which began in 2011/2012 and ended one year later). While the expression of *GATA-3*, *T-bet* and *TGF-ß* was induced in the muscles followed by an injection of the respective plasmid vectors, the expression of *IL-4/13A*, *IFN-γ* and *IL-17A* in the muscles, spleen and head kidney was variable and not very well correlated to *T-bet*, *GATA-3* and *TGF-ß* mRNA levels in the muscles. One reason might be the high inter-individual variability with respect to the results of gene expression analysis, which is quite common—probably due to the population structure of aqua cultured fish [37]. This will affect statistical calculations based on individual results and may affect the interpretation of such results. In addition, the accuracy of the results depends on the number of fish sampled, the number of replicates and the variance of the responses among individual fish and between fish tanks [38]. The differences observed with respect to the levels of *T-bet*, *GATA-3* and *TGF-ß* mRNA between the two experimental groups (Figure 4 and Figure 5) might be due to variations between individual fish and replicate tanks.

The injection of *T-bet*, *GATA-3* and *TGF-ß* expression plasmids induced the significant muscle tissue expression of the corresponding genes evaluated by qPCR, with the highest expression taking place on days 2–7 post-injection. The expression of mRNAs or transgenes at the injection site following i.m. injection has been observed in numerous studies using different plasmid vaccine vectors and reporter gene expression vectors [39,40,41]. After the re-distribution of expression vectors into the circulation and other tissues and organs, one may find recombinant gene expression distally, such as in the spleen and kidney [42,43,44,45]. Contrastingly, in the current study, there was no upregulation, except at early timepoints, in the expression of *GATA-3* and *T-bet* in the spleen and head kidney. The expression levels of all the genes studied were, on many occasions, higher in the control fish injected with the control plasmid (spleen and head kidney) before and after experimental challenge. We cannot offer any explanations for this, but we would like to propose some suggestions. It has been proposed that *T-bet* and *GATA-3* and their DNA-binding capabilities are partly under epigenetic control [46,47]. Whether salmon possess similar epigenetic modification processes is not yet known. In general, gene expression profiling might be used only as an indicator of functional response, and the interpretation of mRNA levels should, as such, be treated with caution.

The cellular expression and localization of *GATA-3*, *T-bet* and *TGF-ß* in the muscle cells correlated with the in vitro expression study using CHSE-214 cells, where the transcription factor *T-bet* and *GATA-3* was expressed in the nuclei. However, there was a discrepancy related to where *TGF-ß* was expressed in the cytoplasm of CHSE-214 cells, but not in the cytoplasm of muscle cells. Interestingly, the adjacently localized cells surrounding the muscle cells and cells localized along the needle trajectory contained bright and specific fluorescence, suggesting that cells other than muscle cells were expressing the RFP tracer. Similar findings have not yet been reported in fish or in mammals. Following the injection of plasmid vectors into fish muscles, it has been shown that there is a solid influx of inflammatory cells [48]. The *CMV* promoter used in the current study has been shown by others to facilitate gene expression in several cell lines and tissues in fish [49,50,51]. As such, the promoter is not tissue- or cell-specific per se. We are afraid that we cannot specify the type of cells that contain the RFP; this would require, e.g., immunohistochemical work, together with flow cytometry, and we lack the appropriate tools for the identification of cells.

It is acknowledged that fish share the same basic immune cell distribution as many mammalian species, though differences exist [52]. The number of CD3ɛ^+^ cells (presumably T cells) in the spleens and head kidneys of salmonids are much lower compared to the thymus, gills and intestines [53]. In the current study, we found very low expression of *T-bet* in the spleen. This is in contrast to another study, where the expression of *T-bet* was highest in the spleen of non-stimulated fish, but not in the spleen of *A. salmonicida*-infected fish. From other observations, it appears that the expression of *T-bet* in the spleen was significantly regulated during immune stimulation [54,55]. The tissue specific expression pattern of rainbow trout *IL-17A/F2* has been shown to be similar to the distribution of CD3ɛ-positive cells [53], where the intestine, gills and thymus contain higher amounts compared to the rainbow trout spleen and head kidney [56]. It has been shown that, following challenge with *V. anguillarum*, the spleen is the first tissue to sequester the bacteria. We speculate that the reason why the expression levels of *IL-17A* increased so much in the spleen is due to the mode of bacterial infection, where *V. anguillarum* in the earliest phase is sequestered by the spleen [57] and may induce a local immune response similar to the expression pattern of I*L-4/13*. However, the modulated expressions of *IL-17A* and *IL-4/13* were not statistically significant compared to the controls. High individual variations (e.g., high responders) with respect to the level of gene expression may also be a plausible reason for the increased expression in fish that received the *TGF-ß*-encoding plasmid.

With regards to whether the overexpression of *T-bet* may, in theory, induce a certain protection against intracellular pathogens [9,58], we challenged salmon with *P. salmonis*, which is an intracellular bacterium that causes disease in salmon. Likewise, a Th2 response, governed by the overexpression of *GATA-3*, may give a certain amount of protection against the extracellular bacterium *V. anguillarum.* The overexpression of *TGF-ß* might also induce a certain amount of resistance against *V. anguillarum* [58,59]. A correlation between overexpression and protection was not observed, as the experimental challenges of salmon with *P. salmonis* and *V. anguillarum* resulted in mass mortality, where no significant differences, in terms of the percentage of fish that survived, were observed among the different treatment groups. The challenge by *V. anguillarum* produced a rather immediate mortality, at a rate of 100%, even though that only nine challenge bacteria were injected into the fish. In the pilot experiment, 10^3^ bacteria were injected into each fish, resulting in 100% mortality. Based on this result, we believe that, by reducing the dose 100-fold, we could reach a mortality rate below 100%, and possibly observe the effects of the plasmid injections.

The use of plasmid vectors encoding *T-bet*, *GATA-3* and *TGF-ß* in the current study may be exploited further in vaccine trials, where increased protection is needed [60]. This is more in line with the activity of other cytokine-encoding plasmids, such as *IL-15* and *IL-23,* as molecular adjuvants, which induce protection along with specific antibodies in immunization trials followed by challenge. However, such molecular adjuvants alone will not provide complete protection during challenge experiments [61,62].

In summary, our results indicate that the overexpression of *T-bet*, *GATA-3* and *TGF-ß* induced Th1, Th1 and Th17-like responses at the injection site, but not in the spleen and head kidney. The overexpression did not confer any disease protection against *P. salmonis* and *V. anguillarum*. Such overexpression studies might be further exploited to study T cell differentiation and dynamics in fish. Interestingly, both the muscle cells and adjacent leucocytes expressed the transgenes after the i.m. injection of the plasmids.

## Figures and Tables

**Figure 1 biology-09-00082-f001:**
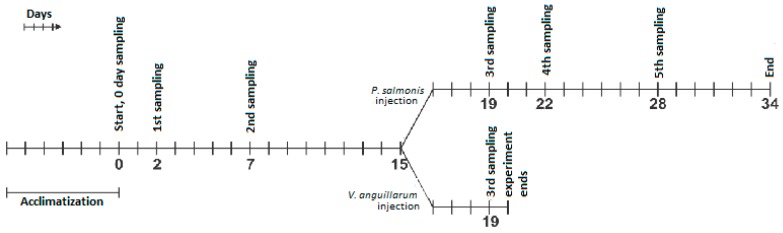
Schematic representation of the experimental plan in this study.

**Figure 2 biology-09-00082-f002:**
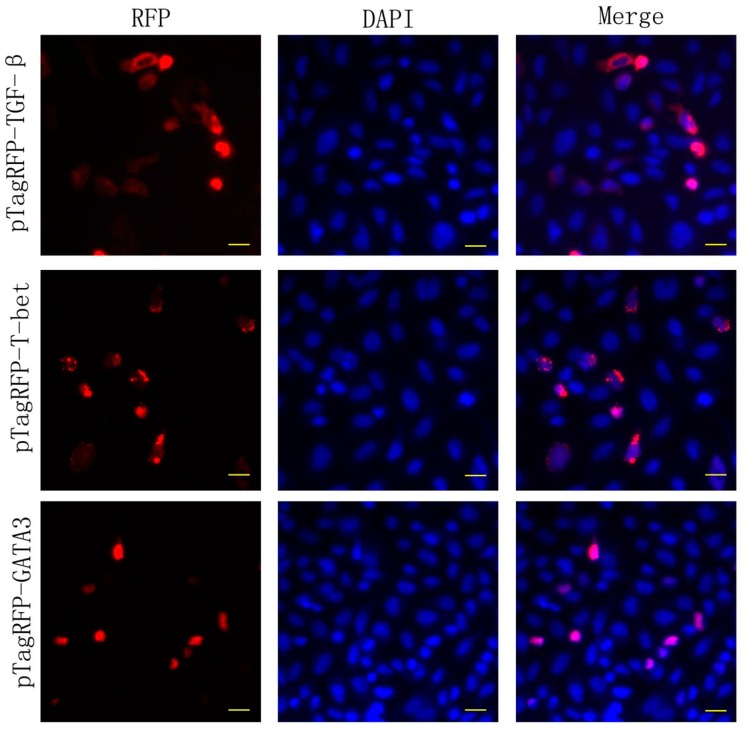
Subcellular localization of recombinant salmon TGF-β, *T-bet* and *GATA-3* in Chinook salmon embryonic (CHSE-214) cells. CHSE-214 cells were transfected with *pTagRFP-TGF-β*, *pTagRFP-T-bet* or *pTagRFP-GATA-3*. After 48 h the cells were fixed in 4% paraformaldehyde and nuclei were stained with DAPI (blue). Bar: 10 μm.

**Figure 3 biology-09-00082-f003:**
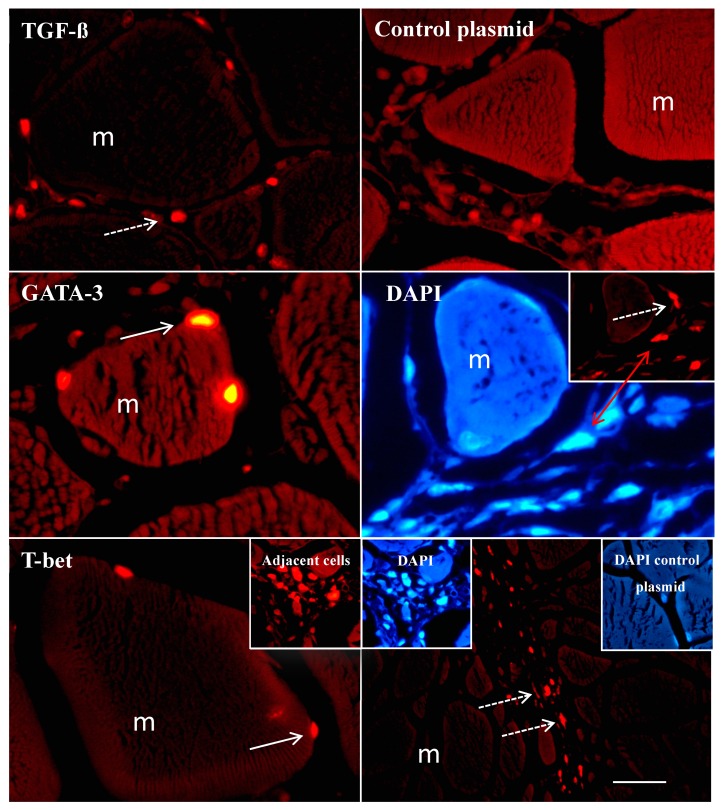
Histological detection of the reporter gene (RFP) from *pTagRFP-TGF-ß*, *pTagRFP-GATA-3* and *pTagRFP-T-bet*-injected fish (muscle sections). Fish were injected with the *RFP*-encoding plasmids or control plasmid. On day 7, the muscle tissue was collected at the injection site and processed for fluorescence microscopy (400× magnification). Bright fluorescence due to the RFP was observed in the nuclei of muscle cells (m) in fish that were injected with *pTagRFP-GATA-3* and *pTagRFP-T-bet* plasmids (continuous arrows), but not in muscle cells in fish that received *pTagRFP-TGF-ß*. Notably, fluorescence was also found in cells surrounding the muscle cells at the site of injections (dashed arrows). The DAPI-stained sections showed nuclear staining (light blue color); in some instances, there was co-localization of RFP (*pTagRFP-GATA-3*) and DAPI in the inflammatory cells (red double arrow). Three fish from each treatment group were analyzed. Bar: 50 µm.

**Figure 4 biology-09-00082-f004:**
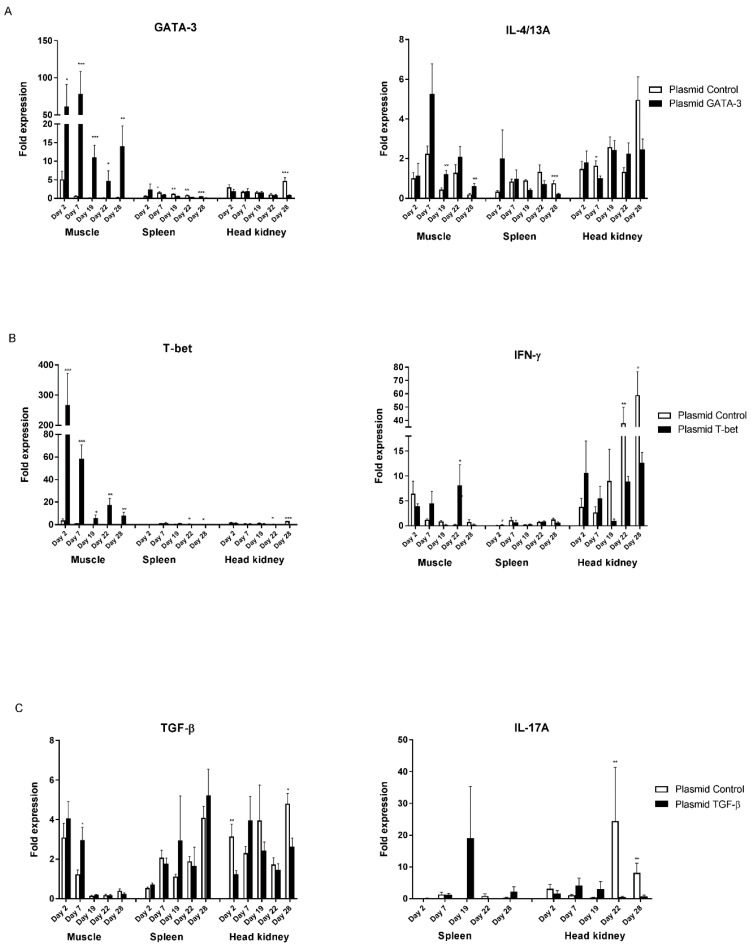
Tissue specific expression at different timepoints after intra-muscular injection and *P. salmonis* challenge. Treated and control fish were injected intraperitoneally (i.p.) with *P. salmonis* (100 µL/fish) on day 15. The bars represent the relative expression levels of (**A**) *GATA-3* and *IL-4/13A*, (**B**) *T-bet* and *IFN-γ*, (**C**) *TGF-ß* and *IL-17A* normalized to elongation factor 1α (*EF-1α*). Each value represents the mean ± S.E.M. (N = 6). Statistical differences compared to their respective controls (*p* < 0.05, *p* < 0.01, and *p* < 0.001) between groups are denoted by asterisks (*, **, and ***, respectively) above the bars.

**Figure 5 biology-09-00082-f005:**
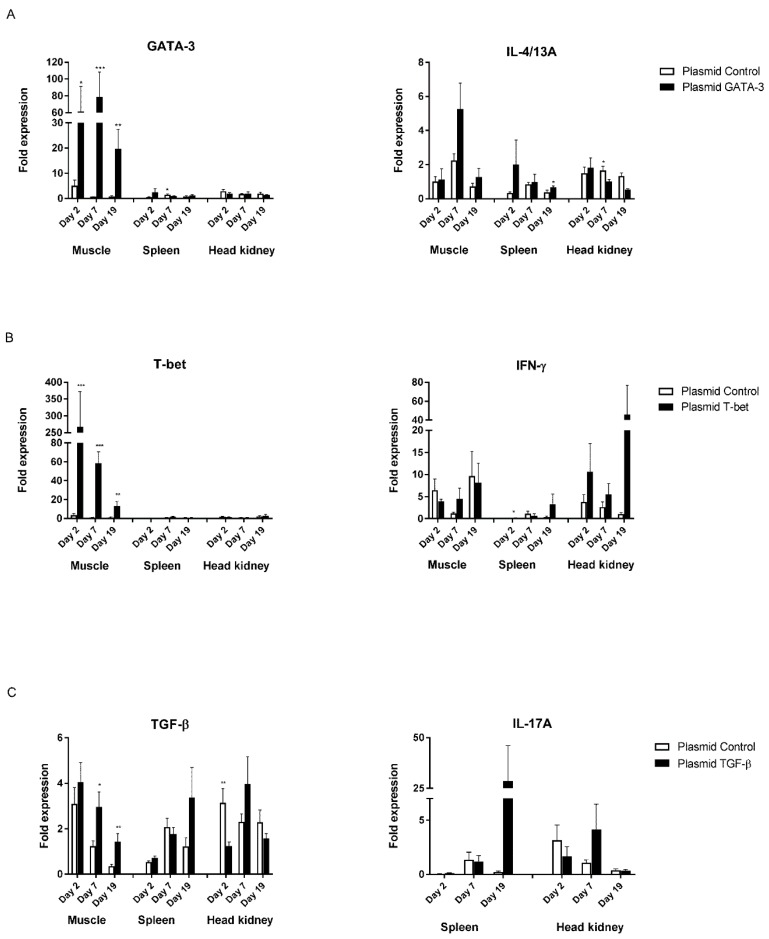
Tissue specific expression at different timepoints post-intra-muscular injection and *V. anguillarun* challenge. Treated and control fish were injected i.p. with *V. anguillarum* (i.p. 10^2^ cfu/fish) on day 15. The bars represent the relative expression levels of (**A**) *GATA-3* and *IL-4/13A*, (**B**) *T-bet* and *IFN-γ*, (**C**) *TGF-ß* and *IL-17A* normalized to *EF-1α*. Each value represents the mean ± S.E.M. (N = 6). Statistical differences compared to their respective controls (*p* < 0.05, *p* < 0.01, and *p* < 0.001) between groups are denoted by asterisks (*, **, and ***, respectively) above the bars.

**Figure 6 biology-09-00082-f006:**
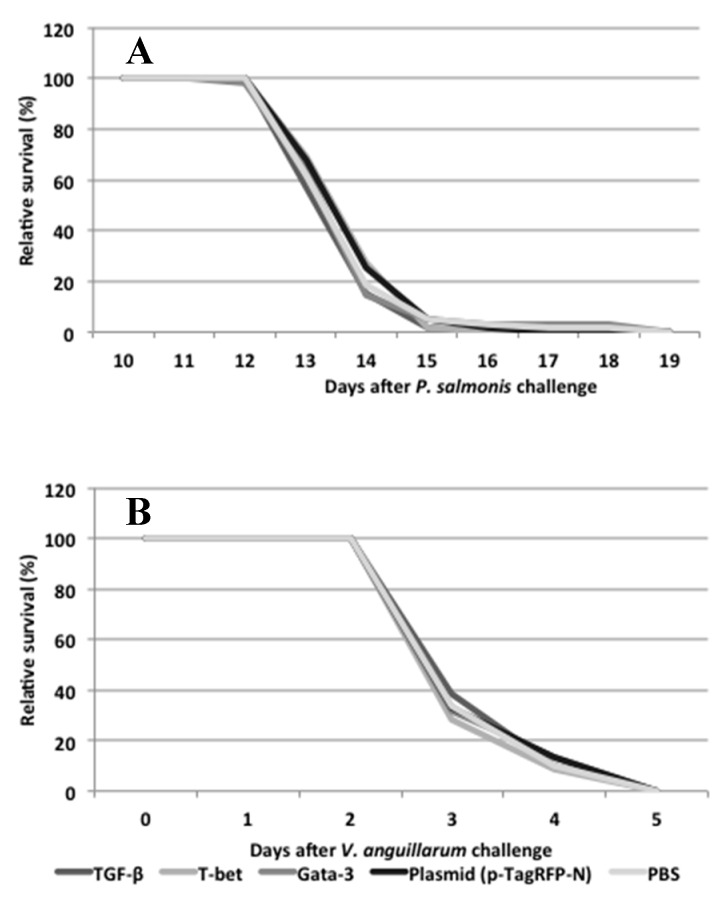
Relative percent cumulative survival (RPS) after experimental infection with (**A**) *P. salmonis* and (**B**) *V. anguillarum*. (N = 60).

**Figure 7 biology-09-00082-f007:**
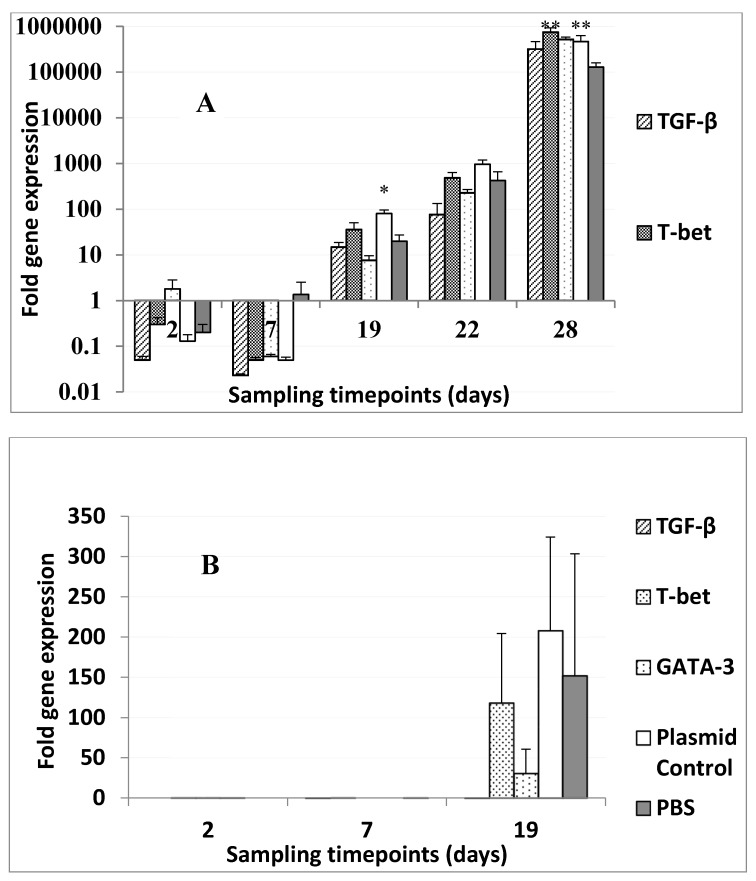
(**A**) Expression of *16S* rRNA *P. salmonis* (**A**) and *V. anguillarum* and (**B**) in the head kidney before and after bacterial infection (day 15) in different plasmid-injected groups. Bars represent the relative expression levels *16S* rRNA normalized to *EF-1α*. Each value represents the mean ± S.D. (N = 6). Statistical differences, compared to their respective controls (*p* < 0.05 and *p* < 0.01), between groups are denoted by asterisks (* and **, respectively).

**Table 1 biology-09-00082-t001:** Distribution of fish in the different study groups.

Group	Injected with	No. Fish in Total	No. Fish for Sampling	No. Fish for Challenge
1	*pTagRFP-T-bet*	210	90	120
2	*pTagRFP-GATA-3*	210	90	120
3	*pTagRFP-TGF-β*	210	90	120
4	*pTagRFP-N*	210	90	120
5	PBS	210	90	120
6	Day 0 sampling	15		
	Total fish	1065	450	600

**Table 2 biology-09-00082-t002:** List of primers and their designated applications.

Gene	Primer	Sequence	Application	Acc. No.
*T-bet*	T-bet _F	TCAGATCTCGAGATGGGCGGCATAGGTGGCAATCTTT	Plasmid construct	GU979861
	T-bet_R	CCGGGCCCGCGGTCAGTGGGAATAAAAGCCGTAGTAG		
*GATA-3*	GATA-3_F	TCAGATCTCGAGATGGAAGTATCCGCCGACCAACCCC	Plasmid construct	EU418015
	GATA-3_R	CCGGGCCCGCGGCTAGCCCATGGCAGAGACCATACTG		
*pTag-RFP-N* vector	pTag-RFP-N_F	ACAACTCCGCCCCATTGACGCAAAT	Plasmid construct	Cat. # FP142
	pTAG-RFP-N_R	CCGCCCTCGACCACCTTGATTCTCATG		
*T-bet*	As*T-bet*_F	CAGCAAAGTGTCACCTCCAA	Real-time	GU979861
	As*T-bet*_R	GGGCTTGTAGAAGCTGTTGC		
*GATA-3*	AsGATA-3_F	CCCAAGCGACGACTGTCT	Real-time	EU418015
	AsGATA-3_R	TCGTTTGACAGTTTGCACATGATG		
*IL-4/13A*	AsIL4/13A_F	CCGACATCTGAGGGTTTACAAC	Real-time	AB574339
	AsIL4/13A_R	TGCCCTCCGCCTGGTTGTC		
*IFN-γ*	AsIFN-γ_F	CGTGTATCGGAGTATCTTCAACCA	Real-time	AY795563
	AsIFN-γ_R	CTCCTGAACCTTCCCCTTGAC		
*V. anguillarum*	Vang16SRNA_F	CATGGCTCAGATTGAACGCTG	Real-time	X71830
	Vang16SRNA_R	CCACATCAGGGCAATTTCCTAG		
*P. salmonis*	Pisci16SRNA_F	AGGGAGACTGCCGGTGATA	Real-time	PSU36941
	Pisci16SRNA_R	ACTACGAGGCGCTTTCTCA		
*EF-1α*	AsEF1α_F	CACCACCGGCCATCTGATCTACAA	Real-time	AF321836
	AsEF1α_R	CACCACCGGCCATCTGATCTACAA		

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
