# Peer review of "Overexpression of T-bet, GATA-3 and TGF-ß Induces IFN-γ, IL-4/13A, and IL-17A Expression in Atlantic Salmon"

_biology, 2020, doi:10.3390/biology9040082_

Round 1
Reviewer 1 Report
The manuscript entitled “Overexpression of T-bet, GATA-3 and TGF-ß induces IFN-γ, IL- 4/13A, and IL-17 expression in Atlantic salmon” described the effect of the intramuscular injection of the expression plasmids encoding for T-bet, GATA-3 and TGF-b in the expression of different T cells polarization markers and in the protection against two bacterial pathogens. In general terms, the manuscript is well written and could be of interest for the fish immunologist. However, some minor questions should be addressed:
- The style used for the murine mutant lines should be consistent (e.g. T-bet−/− mice vs. GATA-3-/- mice)
- The teleost gene and protein names should follow the official nomenclature style stablished for zebrafish genes and proteins (that nomenclature is now being applied to the other teleost species).
https://wiki.zfin.org/display/general/ZFIN+Zebrafish+Nomenclature+Guidelines
- Line 136: Table 2 should be replaced by Table 1 to follow a logical order
- Line 165: and TGF-b??
- Line 204: I think “five days” should be replaced by “four days” to be coherent with the information contained in line 206 and Figure 1.
- The authors affirm that T-bet expression in CHSE-214 cells is mainly detected in the nuclei (line 264) but the corresponding image does not reflect that. The red signal is mainly observed in the cytoplasm.
- Figure 6: I think it could be better to represent the different treatments as colored lines.
- Figure 7: I think it could be better to divide the figure in panel (a) and panel (b), and indicate this in the text. Moreover, the Figure legend is incomplete for the first panel.
- The discussion is poorly elaborated. The authors should improve this section and highlight the results. Moreover, it seems the authors are trying to justify all the time the absence of certain significant results.
Author Response
We thank the referee for many valuable comments. We have now revised the manuscript following the reviwer´s comments, recommendation and suggestions.
- The style used for the murine mutant lines should be consistent (e.g. T-bet−/− mice vs. GATA-3-/- mice)
- Response: We have corrected the text to be consistent throughout the manuscript
- The teleost gene and protein names should follow the official nomenclature style stablished for zebrafish genes and proteins (that nomenclature is now being applied to the other teleost species).
- Response: We have given all gene names in Italics, as pointed out.
- Line 136: Table 2 should be replaced by Table 1 to follow a logical order
- Response: We have inserted Table 2 in the QPCR section. This is more logical than placing it in the end of manuscript. But, we have not replaced Table 1 with Table 2 since we think Table 1 offer the readers a overview of different fish groups.
- Line 165: and TGF-b??
- Response: Done
- Line 204: I think “five days” should be replaced by “four days” to be coherent with the information contained in line 206 and Figure 1.
- Response: Done
- The authors affirm that T-bet expression in CHSE-214 cells is mainly detected in the nuclei (line 264) but the corresponding image does not reflect that. The red signal is mainly observed in the cytoplasm.
- Response: We do not agree. The figure 2 (pTagRFP-T-bet; middle panel) clearly shows nuclei co-localization of RFP with DAPI.
- Figure 6: I think it could be better to represent the different treatments as colored lines.
- Response: The figure shows that there is no disease protection governed by the plasmids. Hence, it is no major point to the lines their individual colors. We now have tried to color the lines, but they appear messier than the present ones.
- Figure 7: I think it could be better to divide the figure in panel (a) and panel (b), and indicate this in the text. Moreover, the Figure legend is incomplete for the first panel.
- Response: We have done as suggested. We have also moved text to appear in a more logical in accordance with the figure.
- The discussion is poorly elaborated. The authors should improve this section and highlight the results. Moreover, it seems the authors are trying to justify all the time the absence of certain significant results.
- Response: We have deleted some speculations, this is in accordance with the reviewer’s comment to avoid justification of absence of results.
Reviewer 2 Report
This paper is well written, with the background, methods and materials, and honest conclusions clearly communicated. All in all, it was a pleasure to read.
Perhaps one of the most important features of this article is that even if well thought out experiments are undertaken, the outcome is the honest truth that there is no correlation with increasing the expression of these transcription factors and increased survival in pathogen-challenge trials. This solidifies what we immunologists know very well -- that disease resistance/susceptibility is very complicated, redundant in many situations, and mostly genetic. Future studies by this group should extend this work to different genetic strains of fish. Some discussion on the impact of genetics on the data would be helpful.
It was interesting to see that even the empty expression vector had an effect in several situations.....and probably due to TLR, or other pathogen pattern recognition pathway. The authors did not comment on this topic.
The authors should comment on whether or not their LD50 levels for V. ang were correct -- afterall, there were very high mortality levels, and apparently even in the controls.
But again, the main value of this publication is in the methods and materials for others to follow.
Author Response
Many thanks for the comments. We appreciated these!
Points to consider:
1. Some discussion on the impact of genetics on the data would be helpful.
Response: Reviewer 1 thought that we included too much speculations to explain absence of results. So we deleted a few of them to make the discussion more to-the-points. We are afraid that by including breeding and genetics we reintroduce more speculations. Thus, we would like not to include breeding/genetics.
2. It was interesting to see that even the empty expression vector had an effect in several situations.....and probably due to TLR, or other pathogen pattern recognition pathway. The authors did not comment on this topic.
Response: The backbone (commercial plasmid) of all recombinant plasmids were identical, so the differential contribution (between the different plasmids) giving TLR receptor mediated signalling was considered to be minor. Thus, we choose not to discuss this issue further. We hope this is satisfactory.
3. The authors should comment on whether or not their LD50 levels for V. ang were correct -- afterall, there were very high mortality levels, and apparently even in the controls.
Response: Good point. We have included a brief discussion on this issue.
"The challenge by V. anguillarum gave quite immediate and 100% mortality – even 9 challenge bacteria was injected into fish. In the pilot experiment 103 bacteria were injected into each fish resulting in 100% mortality. From this we believed that by reducing the dose hundred-fold we could reach mortality below 100% and possibly observe effects from plasmid injections".